# Heat Health Prevention Measures and Adaptation in Older Populations—A Systematic Review

**DOI:** 10.3390/ijerph16224370

**Published:** 2019-11-08

**Authors:** An Vu, Shannon Rutherford, Dung Phung

**Affiliations:** School of Medicine, Gold Coast Campus, Griffith University, QLD 4222, Australia; d.phung@griffith.edu.au

**Keywords:** heatwaves, older people, vulnerable population, heat action plans, prevention, adaptation, Ottawa charter

## Abstract

The population of older people is increasing at a rapid rate, with those 80 years and older set to triple by 2050. This systematic review aimed to examine older people’s perceptions and behaviours against existing heatwaves prevention measures and systematically categorize and analyse those measures using the Ottawa charter for health promotion framework. Peer-reviewed published literature between 22nd September 2006 and 24th April 2018 was retrieved, according to the PRISMA guidelines, from five different databases. Eighteen articles were finally included. There is a lack of published studies from developing countries. Results were categorized and analysed using the Ottawa charter five action areas. Mitigation strategies from current heat action plans are discussed and gaps are highlighted. A lack of systematic evaluation of heat action plans efficacy was identified. Older people are not demonstrating all recommended preventative measures during heatwaves. Support personnel and health professionals are not being pro-active enough in facilitating prevention of adverse effects from heatwaves. Governments are beginning to implement policy changes, but other recommended support measures outlined in the Ottawa charter are still lacking, and hence require further action. Linkage between specific components of heat action plans and outcomes cannot be ascertained; therefore, more systematic evaluation is needed.

## 1. Introduction

The intergovernmental panel on climate change [IPCC] has reported with medium to high confidence that longer lasting and more intense increases in ambient temperature or heatwaves will continue to cause adverse impacts on human health across the globe [1]. Heatwaves, as a consequence of human-induced climate change are widely acknowledged amongst the scientific community as a real and present threat to human health [1]. While the detrimental impacts of climate change continue to transcend the trajectory of time, natural ecosystems and human species have limited capacity to adapt to such changes [2]. Historical examples occurred in Chicago in 1995, across Europe in 2003, Australia in 2009, and Russia in 2010, resulting in a total of over 82,000 excess deaths from heatwaves with older people over-represented [3,4,5,6,7,8]. Consequently, increasing morbidity and mortality from extreme temperatures is becoming a major public health concern for communities worldwide, especially amongst vulnerable groups [9].

Studies on heat-related morbidity and mortality have consistently identified subgroups within the population that are more vulnerable to heat, with a disproportionate percentage of older people to be influenced by these extreme temperature changes [10,11]. Much is known of the physiological effects of heat stress, with many studies linking extreme heat to increasing morbidity and mortality [12]. Older people have less capacity to adapt to heat stress due to natural physiological decline and impaired thermoregulatory functions [13]. In addition, older people also tend to suffer more from chronic diseases compared with younger age groups [14].

Research has shown that those suffering from chronic diseases, such as diabetes and cardiovascular or respiratory illnesses, have even less capacity to adapt to sudden fluctuations in temperature during a heatwave [13]. Oudin Åström et al. [10] reported a strong correlation between increasing mortality rates amongst older people already suffering from cardiorespiratory illnesses. Moreover, Yu et al. [15] have pinpointed a relationship of 2–5% increase in all-cause mortality per 1 °C increase in temperature. Medication dependency further compounds the adverse health risks from extreme heat [16]. This scenario applies to older people worldwide [10,17].

Longer life expectancies and falling fertility rates means that the global population of older people continues to grow at a steady rate, with those 60 years and over surpassing all groups under 60, estimated at 2.1 billion by 2050 and 3.1 billion by 2100 [18]. In 2018, those aged 65 and over outnumbered children under five years of age. The population of older people aged 80 years and over is expected to triple to 426 million by 2050 from 143 million currently [19].

Age, coupled with pre-existing illnesses and several other extrinsic and intrinsic factors, contributes to the overall vulnerability of this population [20]. Factors such as low socio-economic status, ethnicity, gender, lack of education, living alone, or infrequent social outings have all been linked to higher risks of heat-related deaths [3,21,22,23,24,25]. Therefore, it is paramount that measures are in place to mitigate the effects of heat vulnerability and the resultant burden of disease in this population.

This systematic review aims to contribute to the current body of research on factors influencing older people’s adaptive behavior and current preventative strategies during extreme heat events, with two objectives: (i) investigate older people’s thoughts and behaviors in relation to current mitigation measures and their evaluations, in preventing adverse outcomes of heatwaves, and (ii) systematically categorize and analyze those findings by using a health promotion framework.

### Ottawa Charter for Health Promotion Framework

The gradient of health to illness corresponds to a social gradient of high to low socio-economic status [26]. Ultimately, the social determinants of health are the conditions and environment in which people are born, grow, live, work, love, and age [27]. This systematic review defines ‘heat prevention, adaptation or mitigation measures’ as actions taken by the individuals, as well as local communities and federal agencies as specified by regional legislations and policies, hence the use of the Ottawa charter as a framework. It is indeed essential to examine older people as target populations, but also their thoughts and behaviors so as to inform implementation and/or refinement of policies and procedures catered for such populations.

On the basis of this, it is appropriate to utilize the Ottawa charter as a framework to identify inequities and inequalities influencing the health outcomes of older populations in the face of climate change-induced extreme heat events. Initially proposed as a means to promote individuals’ control and improvement of their health, adoption of the framework is correlated with positive outcomes of health promotion programs [28]. This framework comprises of five areas for action: 1. Build healthy public policy, 2. Create supportive environments, 3. Strengthen community actions, 4. Develop personal skills, and 5. Reorient health services.

Build healthy public policy: Optimal population health begins with responsible management [27]. Therefore, governments’ fiscal policies and legislation require multi-sectoral input and appropriate prioritization while taking into account social policies to ensure greater equity. This leads to healthier environments in which goods and services are accessible to all. Subsequently, barriers to adopting healthy public policies are lifted, ensuring easier access to health even from non-health sectors. Thus, the healthier choice becomes the natural choice for both consumers and policy makers alike [27].

Create supportive environments: The socio-ecological model of health highlights the interrelatedness and complexities of individuals within societies and their inseparable connections to their surroundings [27]. On the basis of this advances the notion of sustainability and conservation of our natural resources. Accordingly, compromises must be made to ensure harmonious integration of the built environments while maintaining the integrity of natural surroundings. Only by conserving finite natural resources in rapidly changing environments can we facilitate safe and stimulating living and working conditions that are satisfying, enjoyable, and conducive to health [27].

Strengthen community actions: Empowering communities to take charge of their own fates through collective decision making, planning, and execution of contextually relevant priorities would ensure healthier lives as a whole [27]. Consequently, it is of paramount importance that those communities are given the means to help themselves as well as providing support for others, which in turn facilitates the growth of social capital and strengthens the support networks within, benefiting them both individually and collectively [27].

Develop personal skills: Individuals require a repertoire of knowledge and skills to draw upon in selecting the best choices conducive to and in control of their health [27]. These choices take into account the individuals’ personal circumstances, whether physical, psychological, or socioeconomic, in relation to their immediate environments. Therefore, it is essential that people are provided with appropriate information to enhance their health literacy and life skills through various formal and informal institutions, including educational, professional, commercial, and voluntary bodies [27].

Reorient health services: In this globalized world it is crucial that healthcare systems deliver culturally sensitive and contextually sound services [27]. Those systems must also consider the needs of each individual as a whole in the pursuit of community health. Subsequently, those learning healthcare systems require a concerted effort that transcends various socio-political institutions as well as private and public sectors, resulting in a multi-sectoral wholesome collaboration that is beyond purely clinical and curative services [28,29].

## 2. Methods

This review followed the preferred reporting items for systematic reviews and meta-analyses (PRISMA) checklist for systematic reviews [30]. 

### 2.1. Specifications for Systematic Review

The research question and evidence collection was formulated according to four elements: population, intervention, comparison and outcomes (PICO) (see Table 1).

### 2.2. Search Strategy

Online databases were searched, including the cumulative index of nursing and allied health literature (CINHAL), medical literature analysis and retrieval system online (MEDLINE), PubMed, excerpta medica database (EMBASE), and web of science. All studies since inception to July 2019 were searched using Boolean logic and key words as follows: (Heat OR heat-related OR “high temperatures” OR heatwaves) AND (Elderly OR older people OR seniors OR “vulnerable populations” OR “vulnerable groups”) AND (Management OR “preventative measures” OR prevention OR solutions OR mitigation OR adaptation OR intervention OR evaluation) NOT (Occupation OR work). Duplicates were removed in EndNote library and the three-tier approach was utilized—article titles, abstracts, and full-texts were screened against inclusion and exclusion criteria.

### 2.3. Inclusion and Exclusion Criteria

Peer-reviewed, full-text, qualitative, and/or quantitative studies in English of any design/methodology were considered. Those investigating or evaluating vulnerability and/or adaptability of older people 65 years and older living independently, against a heat action plan (HAP) or preventative measures, were included in this study. Conference abstracts or reviews/editorials, non-English, full-text articles, and grey literature were excluded.

### 2.4. Quality Assessment

Due to heterogeneity of methodology and designs of included studies, it was not possible to apply stringent guidelines in quality assessment.

### 2.5. Data Analysis and Synthesis

Included studies were analyzed and data were extracted onto a spreadsheet under key headings author/year, country, objectives, subjects, study design, intervention/factors, outcomes, and recommendations. A narrative approach was undertaken to synthesize the results since a quantitative meta-analysis was impossible to perform, due to studies’ heterogeneity in methodology, designs, definitions of exposure, and outcomes. To structure the analysis, this review adopted a novel approach by grouping the findings into categories and guided discussion using the Ottawa charter for health promotion framework [27].

## 3. Results

A total of 7534 articles were found from electronic database searching. These were imported into the EndNote library and duplicates were removed, leaving 7133 articles. Title screening eliminated a further 6513 articles. Altogether, 620 articles underwent further abstract screening and 30 articles were selected for full-text screening. Of these, one study examined nursing home residents only [31]; one study did not fulfil the minimum age limit of 65 [32]; one article discussed heat susceptibility and health promotion in general but did not conduct any studies or evaluation [33]; one study discussed health promotion but not related to heat health [34]; three studies assessed mortality/morbidity rates related to heat but not against any components of heat action plans [35,36,37]; five articles were not assessing heat health and/or older people specifically [38,39,40,41,42]—all twelve were excluded. A total of 18 articles remained for analysis (See Figure 1).

### 3.1. Study Characteristics

Results of selected studies have been summarized in Table 2. Date of publication ranged from 22nd September 2006 to 24th April 2018. All studies were in developed countries—seven from Australia [43,44,45,46,47,48,49], three from Italy [50,51,52], three from America/Canada [53,54,55], three from the UK [56,57,58], one from Germany [59], and one from Japan [60].

Various designs and methods of analysis were employed. There were two randomized controlled trials (RCTs) [49,60], ten cross-sectional surveys via telephone or online, and/or face-to-face semi-structured interviews or focus groups [43,44,45,46,47,54,56,57,58,59], one logbook/diary survey [55], two quasi-experimental retrospective studies [50,53], and the rest were meta-analyses investigating mortality/morbidity rates against maximum apparent temperatures [51], time-series random effect multivariate evaluating older people’s daily mortality rates pre- and post-introduction of the ‘national heat health prevention program’ [52] and incidence rate ratios of pre- and post-exposure to a prevention program [48].

### 3.2. Heat Warning Systems and Heat Action Plans

The studies gathered described an array of heat warning systems (HWSs) within countries and across the world. All 18 studies outlined different components of heat action plans (HAPs); albeit, some described and analyzed them in more detail than others (e.g., [50,53,54]). In places with established HWSs and HAPs, an array of different meteorological methods can be found in measuring temperatures and the resultant triggering thresholds for public warnings (e.g., [50,51,52,53,54]. It seems that geographical and climatic differences between regions have influenced the methodology and implementation of warning systems. For example, Philadelphia’s HWS (Pennsylvania, USA) is the oldest, compared with Dayton (Ohio, USA), Phoenix (Arizona, USA), and Toronto (Ontario, Canada) [54]. Instrument-wise, Michelozzi et al. [51] described two different models for estimating daily mortality using maximum apparent temperature and oppressive air mass approaches. Moreover, Sheridan [54] revealed that while Dayton issues alerts if one to two deaths are expected, Philadelphia only sends out alerts when four or more deaths are expected.

Monumental heatwaves in 2003 prompted many countries in Europe to implement both formal and informal processes to combat future occurrences. Abrahamson et al. [57] have included a revised 2004 HAP from England’s department of health which outlines protocols for national, regional, and local levels, as well as most-at-risk subpopulations—especially females >75, those with chronic and mental health illnesses, and people living or working in overexposed areas. Michelozzi et al. [51] provided a complete overview of Italy’s department for civil protection and the ministry of health national HWSs and HAPs, including different methodologies to arrive at implementation for each city, such as local bulletin networks, local registries for subgroups, and a real-time surveillance system. The Italian national prevention program has been operational since 2004 and covers 93% of residents 65 years and over in 34 cities [51,52]. Liotta et al.’s study [50] focused more at a local measure in Rome, evaluating the long live the elderly (LLE) program, designed to counteract social isolation in urban residents >75 years of age. In Germany, the German meteorological service issues warnings when temperature exceeds 32 °C on two consecutive days [59] and informal procedures have been in place for over a decade until 2017, when more formal procedures were implemented [61].

Elsewhere, the Canadian Montreal public health department (PHD) implemented a HAP in 2004, as described by [53]. Examples provided includes activation of alerts by the public health department, through to local activities such as provision of light meals and bottled water, to increasing frequency of phone calls and home visits, to vulnerable people in the community. In the USA and Canada, [54] there was no formal found in HAP in Phoenix, an informal HAP in Dayton, and more formalized HAPs in Toronto and Philadelphia. The Combined Health District of Montgomery County is responsible for heat alerts in Dayton, which coordinates with over 150 local governmental and non-governmental agencies to disseminate heat warnings, advice on how to prevent heat stress, as well as operating hotlines and a “buddy system” in which older people are checked on by volunteers and neighbors. Philadelphia has the most extensive and oldest HWS, dating back to 1991. The division of health promotion coordinates education campaigns and heat–health advisories linking thousands of volunteers and organizations, such as the Philadelphia corporation for the aging to at-risk individuals, via media publications and hotlines. The department of public health is responsible for provision of increased staffing, utilities, and services, such as cooling centers, as required. In Toronto, Environment Canada coordinates with Toronto public health to issue heat alerts and disseminate information to a thousand volunteers and agencies [54]. These local groups are responsible for facilitating hotlines, healthcare professionals, opening cooling centers and home visits, as well as street patrols to deliver bottled water. Despite no formal HAP in Phoenix, the local national weather service agency provides the most detailed advisories that local media can disseminate on a voluntary basis.

Australia currently does not have a formalized national HAP. The government of South Australian capital city Adelaide adopted a more formal HWS only after the damaging 2009 heatwave. The Australian Victorian state government has its own heatwave warning protocols [45,48,49]. In Adelaide, the state emergency service coordinates with the bureau of meteorology to activate heatwave alerts and disseminate prevention advice via media announcements. Pre-determined at-risk individuals are registered by family, friends, and treating GPs with daily phone contacts and home visits as required by NGOs, such as the Australian red cross. The Victorian HAP referenced by Hansen et al. [44] is a revised version of the 2009–2010 frameworks, the auditor general’s 2014 account heatwave management: reducing the risk to public health and the state heat plan 2014 combined [62]. It sets forth protocols for pre-season, during, and after a heatwave to evaluate the effectiveness of activities undertaken. The department of health and human services chief health officer coordinates with the emergency management commissioner and the bureau of meteorology to disseminate alerts and detailed advice to the public.

The Japanese HWS was developed in 2006 under the management of the ministry of the environment and the ministry of health, labor, and welfare [60,63]. Local prevention activities rely heavily on volunteer groups known as ‘Minsei-iin’ or welfare commissioner committees [60].

### 3.3. Risk Factors Awareness, Perception of Vulnerability, and Protective Behaviors

Factors such as age, low socio-economic status, living alone, lack of social outings, use of mobility aids, comorbidities, and use of medications have all been identified as risk factors during heatwaves [43,49,50,53,55]. In general, the majority of subjects were aware of physical health risks but less so of social risks [58]. Healthcare professionals displayed the most awareness of their patients’ risks while community workers displayed less, with crucial knowledge gaps identified regarding specific mechanisms of thermoregulation and safe use of electric fans [45,46,60]. Notwithstanding knowledge levels, in four studies from Australia, UK, and Germany, frontline healthcare professionals appeared to be too complacent and not proactive in seeking out the most vulnerable older people during heatwaves [43,46,56,59].

Despite awareness of danger, many older people surveyed in Australia, UK, and USA/Canada did not feel they were susceptible to heat-related illness [54,57,58]. Many respondents were not concerned upon hearing heatwave warnings, with 74% reporting that they should not have to stop daily activities, keeping most of their usual appointments, and would still exercise despite the heat [47]. In addition, those nominated by older people as advisors/assistance were not proactive in reaching out to them prior to heat events, but rather waiting for them to call when already distressed, citing reluctance to impingement on their independence [58].

In regard to protective behaviors, respondents called upon ‘common sense’ [56,57]. Most displayed adaptive behaviors such as wearing loose clothing, taking showers, and drinking more water [45,48]. However, the full range of protective behaviors was not demonstrated, as evidenced in White-Newsome et al.’s study in Detroit, USA [55]. In instances where bottled water was delivered with heat–health messages, increased water intake was noted, and alternative cooling methods employed [60]. Indoor temperature influenced behavior up to a threshold with increasing temperature lowering the likelihood of people leaving their home or adopting alternative cooling methods, such as seeking cooling centers [55]. At least a third of respondents expressed concerns regarding costs of running air conditioners [49,54].

### 3.4. Effectiveness and Limitations of Heat Action Plans

The success in mortality and/or morbidity rates reduction and improvement in adaptive behaviors have been attributed to introduction of HAPs [48,51,52,53]. However, the results show that while in general most subjects were aware of heat warnings and advice, many were also confused as to what to do exactly [54]. Some studies have shown evaluation of various components of respective regional HAPs, but direct causal relationships between individual components of HAPs and quantitatively measurable outcomes/behaviors have not been established [48,49,50,52,58,60].

## 4. Discussion

The aim of this review was to gather current evidence on risk factors awareness, perception of vulnerability, and protective behaviours of older people against existing government-issued mitigating measures in the events of heatwaves. This also includes professional and non-professional personnel responsible for the health and welfare of older people. Evidently, several important points are highlighted, including (1) HAPs are heterogeneous, relatively new in development, and lacking formal evaluation; (2) It is difficult to establish causal relationship between HAPs components and outcomes and thus further research and refinement in methodology is required; and (3) Further action is needed to translate knowledge/warnings into heat adaptive behaviours. Consequently, effective and efficient health promotion strategies, involving all sectors of the community is required, transcending organisational and political boundaries [27]. As such, utilization of the Ottawa charter potentially facilitates this process in highlighting gaps and guiding further actions. This framework remains as relevant today as it was first proposed over thirty years ago [28,63].

### 4.1. Ottawa Charter Action Areas

#### 4.1.1. Build Healthy Public Policy

National governments have a responsibility to create a more sustainable and healthier world [64]. As evidenced in this study, HAPs were introduced all over the world only after severe losses of lives, coupling with economic losses [65]. It is encouraging to see that inequities are being addressed through implementation of federal and regional initiatives. For example, the LLE program examined by Liotta et al. [50] presents an example of a social policy which bridges the gap between individuals with high social capital and those without. By increasing social support, the program has also reduced health inequity, since those without financial means are able to receive the same care that otherwise only those with higher income can access.

Toloo et al.’s systematic review [66] also found an overall reduction in mortality rates attributable to implementation of HWSs and HAPs, including one study which estimated the total cost of $210,000 (USD) to run a HWS versus a much less cost-effective figure of $4 million per person in saving 117 lives. Despite this evidence, formal policies and legislations protecting the health of older populations in extreme heat events are still lacking, as identified in this review and reflected in a study in Belgium and The Netherlands [67], especially for socially isolated individuals. Evidently, further action by governments worldwide is urgently required to address this global public health issue as a matter of priority.

#### 4.1.2. Create Supportive Environments

The environment in which people live, their income, housing conditions, and access to healthcare all contribute to their ultimate health outcomes [68]. This study has found the same social risk factors that contributed to the mortality/morbidity outcomes during heatwaves [43,44,58]. Those living in urban areas in high rise apartments, with highly impervious surfaces and low greenspaces, suffered from the heat island effect [55]. People suffering from chronic diseases and those on medications were the ones most at risk [49]. The aging people with lower income who could not afford air conditioning or running costs were most disadvantaged [43,49], as well as those relying on mobility aids that may or may not be living in appropriate housing conditions [47]. In addition, marginalised people in society, such as those who are homeless, are much worse off since they may not have access to basic amenities or healthcare at all [69].

With an estimate of 68% of the world population living in urban areas by 2050 and heatwaves posing a serious threat for the urban population, studies have investigated mitigating measures to adapt the built environment, making them more resilient to heatwaves [70,71,72,73]. A cool retrofitting toolkit proposed by Hatvani-Kovacs and Boland [71] to remodel existing precincts could make them more energy efficient and affordable, while Bennetts et al. [70] proposed a combination of behavior modification and cooling refuges to increase thermal comfort and shielding penetration of heat. Both studies have concluded that there are means to reduce the impact of heatwaves without resorting to excessive energy consumption or high costs [70,71]. Those strategies present alternatives to current practices that policy makers could implement to mitigate the impacts of future heatwaves that are sustainable and equitable for older populations and do not further exacerbate greenhouse gas emissions.

#### 4.1.3. Strengthen Community Actions

Poor social capital from infrequent social outings and contacts has been acknowledged as a risk factor for older people during heatwaves [48]. Empowerment of communities leads to development of strong social capital which would lend support to individuals in need when they need it most. For example, preventative measures during heatwaves would not have operated as well or as efficiently if not for the efforts of the ‘Minsei-iin committees’ or the ‘buddy systems’ described by Takahashi et al. [60] and Sheridan [54]. Further, the importance of recognised NGOs such as red cross cannot be ignored during emergencies [48]. Additional contacts from neighbours, family, and friends also provide essential linkages to improve favourable health outcomes.

However, social contacts and support systems need to be ‘active’ rather than ‘passive’, as described by Wolf et al. [58]. Since older people value their independence and nominated support people are reluctant to impinge on their independence, it is of paramount importance that any preventative measures be implemented in a timely ongoing manner to ensure maximal efficiency and effectiveness [57,58].

Furthermore, studies into community cohesion and healthy aging have found that older people are more likely to participate in health promotion initiatives, improved mental well-being and self-evaluated health if they perceived the neighbourhood to be safe and accessible [74]. Coll-Planas et al.’s [75] study into prevention of social isolation and loneliness in the older people through weekly group activities found that after two years, 39.5% still participated in the activities, almost half maintained contact with each other, and overall they reported improvement in mood and social wellbeing. Similarly, Harada et al.’s study [76], led by Kobe University, found that older people’s social network improved significantly following a year-long, 18 theme-based programs aimed at promoting social interactions. It is the duty of local governments and communities to provide older people with opportunities to build upon their social capital, thus lowering the risk associated with extreme heat events.

#### 4.1.4. Develop Personal Skills

This study has identified different means in which authorities strived to educate their citizens on preventative measures during heatwaves. The two most commonly utilised methods were via television and radio [57]. However, more modern means of communication have also been employed with some degree of success, such as mobile messaging and social media advisories [44]. Evidently, there is much room for improvement in this area however, as approximately 47% of respondents still ignored warning messages [47]. Another issue of concern is confusion or poor recall of exact course of actions and under-utilisation of recommended mitigation activities [54,55].

Baker et al.’s study [77] into health literacy and mortality in older people found that poor health literacy is correlated with poor uptake of preventative services, with an overall strongest correlation between reading fluency and all-cause mortality. Geboers et al.’s study [78] also found the same correlation between low education level and low health literacy. Moving beyond just reading or numeracy skills, Serper et al. [79] noted that for older people, cognitive processing such as memory, processing speed, and inductive reasoning all influence functional health. This could perhaps explain the low rates of heat warnings recall, confusion, and ignorance regarding risks to health and lack of translation to protective behaviours [47,55,57]. Another interesting point to note is the success of Nitschke et al.’s randomised controlled trial in which multiple ‘reminders’, such as laminated cards and fridge magnets, were used to facilitate adaptive behaviours [49].

#### 4.1.5. Reorient Health Services

The evidence gathered in this study highlights the fact that community frontline healthcare services require a more proactive multidisciplinary approach in caring for the older people during heatwaves [45,46,56,59]. Johansson et al.’s study [80] in Sweden found that while over 90% of health professionals have a positive attitude to health promotion within healthcare delivery, in practice there were barriers to implementation. The three most cited barriers to health promotion were large workload, inadequate guidelines, and inexplicit objectives. These barriers echoed the findings of McInnes and Ibrahim’s study [46] which concluded that despite the capacity to promote health during heatwaves, extra resources, coordination, and leadership is required.

However, reorientation of healthcare services is possible, as demonstrated by the ‘The Core Skills in Health Promotion Project’ described by Yeatman and Nove [81]—using the capacity building framework and three key elements of partnership, leadership, and commitment to achieve workforce expansion, reallocation of resources, and organizational change. Another way to reorient health services is through modern wearable technology. The global aging population means an increasing need for more cost-effective, efficient early interventions of chronic diseases [82]. Research on wearable, non-invasive, remote monitoring devices, such as fabric-integrated sensors and removable wrist/body sensors, have predicted much promise into a more integrated, sustainable, and equitable healthcare system [83,84,85]. Older people are more likely to agree to installation of remote monitoring systems in their homes if it means they could continue to live independently for as long as possible, rather than moving to an aged care facility [86,87].

Only by empowering older people with the means to monitor their own health and by facilitating collaboration between them and healthcare providers can we move away from curative services into more preventative and sustainable healthcare systems.

#### 4.1.6. Interrelatedness of Action Areas

All five action areas of the Ottawa charter are connected to one another. Optimal health can only be achieved by combining all aspects of social determinants of health and addressing all intrinsic and extrinsic factors influencing health outcomes. A study by Williams et al. [88] has found a similar network of relationship in factors influencing adaptation to extreme heat (see Figure 2). However, only two studies [50,60] demonstrated utilisation of all action areas of the Ottawa charter with seemingly more favourable outcomes (see Table 3).

Evidently, relying on just one or two action areas of the Ottawa charter is not enough. As aforementioned, older people, support personnel, and frontline healthcare professionals all tend to underestimate the risks and level of vulnerability, similar to findings from a previous study by Bittner and Stobel [89]. Furthermore, a recent review by Wondmagegn et al. [90] has identified a substantial burden on healthcare systems in economic terms attributable to extreme heat exposure. Also, consistent with this review, those most affected were older people, females, as well as those from ethnic minorities and with low socio-economic status [90]. Hence, this reinforces the need for a more comprehensive multi-faceted strategy to derive the most effective solutions in solving this complex, but potentially solvable public health problem.

### 4.2. Limitations

This review focused on only peer-reviewed, English language studies. As such, non-English, unpublished, and grey literature potentially containing relevant informal evaluation of heatwave plans may have been missed. Two studies by Moher et al. [91] and Morrison et al. [92] found no risks of bias from excluding non-English papers. Nevertheless, studies have provided a growing body of evidence that grey literature may provide more detailed evaluation of interventions while not being subjected to publication bias [93,94]. Also, Adams et al. [93] have concluded that results from evaluation of public health interventions may predominantly be held in grey literature, though both studies concluded that the risk of bias is high and replicability of future searches is low when analysing grey literature [93,94].

Though quality assessment of articles in this review was not able to be carried out, individual authors identified their own paper’s limitations. These include respondent bias [45,46,55], selection bias, and attrition bias [49,60], comparability issues between data sets [48], recall bias [44,47,57], low response rate [46], inability to generalize outcome to the rest of the population, or not capturing all sections of a community due to no phone access or non-English speakers [50,54,57].

In addition, the adoption of various study designs from ‘observational’ to ‘experimental’ means that each study has its own strengths and weaknesses [95]. For example, only two studies [49,60] applied RCT design, which is considered as the ‘gold-standard’ of experimental designs as it minimizes selection bias, resulting in less confounded outcomes with clearer causalities [95,96]. However, as aforementioned, those studies were not without confounders, as fully randomized or double-blind designs in medical/health research can be fraught with ethical and practical issues [96]. With the exception of quasi-experimental studies [50] and [53] as a ‘compromise’ from full randomization, the rest of the included studies adopted more observational designs [95]. These study designs are more economical to implement in terms of time and finance but at the same time they are also subject to potential biases, as mentioned by various authors above [95,96]. Moreover, authors of both [95] and [96] have conceded that there is a place and scope for both ‘interventional’ and ‘non-interventional’ studies as they can be complementary, depending on the aim and objectives as well as ease of operation, duration, and expenses.

Overall, there appears to be a gross under-representation of studies from countries with developing economies, as all eighteen articles were from high income developed regions. Yet, studies such as Herold et al. [97]’s have found that each year low income countries have experienced twice the number of hot days compared with high income countries. Notwithstanding the fact that low income countries are responsible for the least portion of global greenhouse gas emissions, they are situated in the most susceptible climatic zones and consequently already suffering disproportionately from global warming compared to developed countries [97,98]. This highlights a large gap in research and policy development that requires urgent attention in line with increasing climate change.

## 5. Conclusions

The aim of this review was to identify current measures in mitigating the adverse effects of extreme heat events in older populations. These include self-reported perceptions of vulnerability and risk awareness, as well as social and behavioural factors affecting heat–health outcomes of older people during heatwaves. This study has established that older people remain vulnerable to heat but that heat action plans are making a difference to mortality and morbidity rates overall. Healthcare personnel, both professional and non-professional, have the capacities to effect change, but more dynamic engagement is required. Furthermore, establishing a cause–effect relationship between specific mechanisms of heat action plans and health outcomes remains a challenge. Nevertheless, by analysing the findings against the backdrop of the Ottawa charter, this review highlights the need for a more coordinated approach to solve this potentially recurring public health dilemma. Further research with more rigorous methodology and designs is required in this field. There is a strong need for further research, particularly in low income countries given the projected increase in extreme weather events relating to climate change and the dearth of research evidence on this topic available for inclusion in this review [99]. Finally, there is an urgent need to evaluate existing heat action plans to identify the scope of components, the effectiveness of components and their combination, and adjust or delete redundancies.

## Figures and Tables

**Figure 1 ijerph-16-04370-f001:**
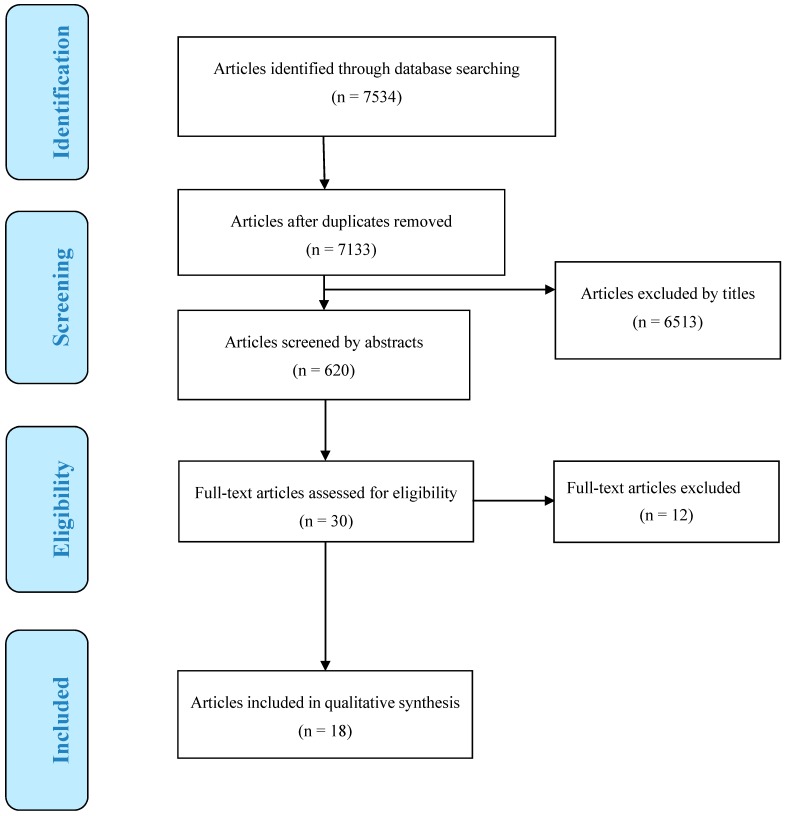
PRISMA flow diagram for literature search.

**Figure 2 ijerph-16-04370-f002:**
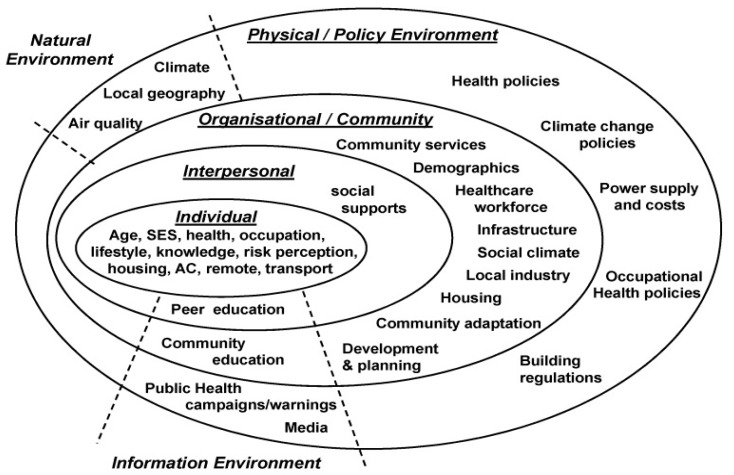
Factors influencing heat adaptation [88].

**Table 1 ijerph-16-04370-t001:** The PICO strategy.

Parameters	Details
Population	Older people 65 years and over
Intervention	Heat prevention, adaptation or mitigation measures
Comparison	No prevention measures
Outcomes	Any observable/measurable effects of heat related mortality or morbidity

**Table 2 ijerph-16-04370-t002:** Summary of selected studies.

Author and Year	Country	Objectives	Subjects	Study Design	Intervention or Factors	Outcomes	Recommendations
**Hansen et al. 2011 [43]**	**Australia**	Investigate heat-susceptibility in older people and perceived adaptation barriers during heatwaves in Adelaide	n = 35 four groups of health providers, managers and policy makers	Telephone interviews and focus groups	Questioning respondents knowledge of risks to older people and barriers to adaptive behaviours	Respondents identified physiological (poor health, chronic conditions, functional disabilities), socioeconomic issues (costs associated with running air-conditioners), psychological issues (anxieties, cognitive dysfunction), and barriers/enablers to adaptive strategies	Clear instructions on operation of air-conditioners Energy rebates for older people Specific strategies for specific medical conditions
**Hansen et al. 2015 [44]**		Investigate prevention behaviours (PB) of independently living residents in South Australia and Victoria	n = 1000 ≥65	Cross-sectional Telephone survey	Demographics, social contacts, self-evaluated health status, coping strategies, medications, air conditioning, and heat warnings	Most demonstrated PB; More heat warnings recall and AC in South Australia vs. Victoria; Female sex, chronic illness sufferers reported increased morbidity	Review current policies Disseminate heat warnings via media and SMS
**Ibrahim et al. 2012 [45]**		Investigate healthcare providers current practices to care for older people living independently in Victoria	n = 327 Six groups	Cross-sectional electronic survey	32 questions - demographic, professional characteristics, heatwave impacts, heat health knowledge, current practices to treat heat-related illness	Most aware of danger to older people; Gaps in knowledge: thermo-regulation, electric fans use and most critical time to offer help; Few emergency plans in place; Reactive and opportunistic in practices	Emergency response plans needed Improvement required in knowledge Call for a more proactive approach
**McInnes et al. 2010 [46]**		Investigate roles of community organisations and health providers in reducing harm to older people living independently in Victoria	n = 12 Four groups	Cross-sectional study, face-to-face and telephone survey	Semi-structured interviews exploring their roles in an heatwave emergency and issues such as coordination, identification of high-risk persons and training/education	No formal heat action plans (HAPs); At-risk individuals identified prior to summer; Good communication networks available, potentially able to provide appropriate care but lacking coordination and training; Mainly reactive and opportunistic activities	Need formalised heat action plans More proactive strategies and practices More resources and training needed Develop ’buddy’ system of volunteers
**Nitschke et al. 2013 [47]**		Investigate resilience, prevention behaviours, risk factors and health outcomes of independently living residents in South Australia	n = 499 ≥65	Cross-sectional computer assisted telephone survey	Survey explored demographics, housing, social connectedness, self-reported health status and vulnerability, heat health knowledge and resilience	Majority are resilient; Variety of prevention behaviours reported; High medication usage for chronic diseases, female sex, mobility aids, chronic diseases, mental health increased risk and poorer outcomes; Less social contact for those <75	Targeted intervention required to address medication use, co-morbidities, knowledge improvement and social isolation Policy development required
**Nitschke et al. 2016 [48]**		Investigate effectiveness of heatwave warning system in Adelaide	Residents of all ages	Comparing morbidity–mortality data ecological design	Incidence rate ratios (IRRs) of daily ambulance call-outs, emergency presentations and mortality data from 2009 and 2014 heatwaves	Significant reduction in morbidity especially emergency presentations in 75+ group; No reduction in mortality rate	In-depth assessment of services provided during heatwave including reach and behaviour change More studies into mortality risks factors
**Nitschke et al. 2017 [49]**		Investigate effectiveness of targeted information in preventing adverse health outcomes during heatwave	n = 637 ≥65	RCT	Intervention group provided with specific instructions on heat protective measures; Control group advised to follow media and seek own medical assistance as needed	Higher use of AC, wet cloth on face/body and significant heat stress reduction in intervention group; Control group also demonstrated protective behaviours through media	Results generalizable to other older people population in SA Further studies on built environment thermal comfort, social services, GPs active involvement
**Liotta et al. 2018 [50]**	**Italy**	Assess effectiveness of long live the elderly (LLE) program in reducing heat-related mortality from social isolation of independently living residents	n = 12207 ≥75	Quasi- experimental retrospective cohort study	Intervention group given social support and all health needs via both formal institutions and volunteers; No extra support for control group; Mean property tax evaluation determined SES	Mortality rate reduced 13% under LLE with 25 deaths averted; LLE indirectly reduced impact of low SES and mortality	Routine assessments of older people and provision of case-specific social services could improve health outcomes during heatwaves
**Michelozzi et al. 2010 [51]**		Analyse current practices and methodologies of the Italian national heat prevention program	93% residents ≥65 across 34 cities	Examine dose– response relationship between mortality and maximum apparent temperature (MAT)	Assessing strengths and limitations of different methods to monitor daily summer mortality in 2008, 2003 and reference period 1995–2002, using Rome and Milan as examples	Mortality (MAT) differed across cities; City-specific warning systems, coordinated central information network, constant modulation of preventative measures major strengths; Specific prevention programs ensured timely mitigation measures; Reduction in mortality rate attributable to prevention strategies	Implement local registries to identify vulnerable individuals - ensures uniform identification At-risk individuals require specific home-care plans Further assessment of heat mitigation plans required
**Schifano et al. 2012 [52]**		Investigating effectiveness of heatwave prevention plans post-2003	Residents ≥65 across 16 cities	Multi-centre time series (1998–2002) vs. (2006–2010) random effect multi-variate meta-analysis	Comparing 16 city-specific daily mortality rates pre and post heat prevention measures, by studying relationships between mortality and maximum apparent temperature	Observable reduction in effects of high temperature on mortality rates attributable to mitigation plans	More attention needed at beginning of summer when populations yet to adapt to heat and prevention activities not yet fully functional, and end of summer when the effect of heat is stronger
**Benmarhnia et al. 2016 [53]**	**America/Canada**	Investigating causal effects heat action plans (HAP) and association with different subgroups	Male vs. Female; ≥65 vs. <65; Education first vs. third tertile	Quasi-experimental retrospective - difference-in-differences approach	Comparing daily mortality rates (2000–2003) and post-HAP introduction (2004–2007)	A reduction in 2.52 deaths per day overall with 2.44 deaths per day less for older people ≥ 65; A 2.48 deaths per day less for low SES group; No differences between genders	Specifically targeting vulnerable population may reduce inequalities between populations More frequent home visits and daily phone calls to more at-risk individuals
**Sheridan 2007 [54]**		Investigate efficacy - four heat warning systems in Dayton, Philadelphia Phoenix, Arizona, Toronto	n = 908 ≥65	Cross-sectional telephone survey	Perception of own vulnerability, knowledge of prevention behaviour and course of action during heatwaves	Most aware of heat warnings but few understood what to do; Only ~ half changed behaviour; Main source of warnings from television and radio	Broadcast specific/easy to understand heat health advisories Address warnings ’blocking out’/confusion Explain safe use of electric fans
**White-Newsome et al. 2011 [55]**		Investigate behaviours and adaptability to increased indoor temperatures and environment in Detroit	n = 29 Aged >65	Cross-sectional survey of volunteered residents	Data collection via hourly activity logs of eight heat-adaptive behaviours	Indoor temperature significantly influenced behaviour; More adaptive behaviours in high-rises and highly impervious areas; Changing clothes, taking additional showers and going outside rarely used	Public health interventions outreach to this vulnerable group to encourage full range of prevention behaviours
**Abrahamson et al. 2009 [56]**	**UK**	Explore frontline healthcare professionals’ risks awareness and support for older people at risk of heatwaves adverse effects and perceived barriers to effective implementation of HAP	n = 109 covering three different socio-economic areas	Semi-structured interviews and focus groups	Awareness of details of HAP; opinions of self and organizations’ ability to identify and prioritize high-risk individuals; barriers and facilitators to effective implementation of HAP	Poor awareness of HAP from health professionals; Summer workloads not prioritised with older people in mind citing complexities and classification of vulnerability and infrequency of heatwaves as barriers	Multidisciplinary approach to interventions recommended Further evaluation of existing practices
**Abrahamson et al. 2009 [57]**		Investigate knowledge, perceptions of heat health risks, and protective behaviours of older people living independently	n = 73 Aged 72–94	Semi-structured interviews	Face-to-face interviews with subjects recommended by GPs	Few respondents considered themselves old or vulnerable or at risk of heat related illness, despite being aware of comorbidities; Most respondents disliked ’nanny state’ approach of intervention	Imbed warnings into favourite TV programs Clear/easy to understand instructions Focus on most ’at-risk’ individuals by health professionals Warn community rather than targeting individuals
**Wolf et al. 2010 [58]**		Investigate older people self-reported vulnerability and subsequent influence on adaptive behaviour	n = 105 Aged 72–94 in Norwich and London	Semi-structured interviews and open-ended questions. Respondents (A) and nominated people (B) to whom they turned to for assistance also interviewed	Perceptions and knowledge of heat risks explored including daily routine, socialisation habits, physical activity, actual/hypothetical behavioural changes in response to heatwaves, barriers to do so, medical conditions and medications, and type of housing.	Most (A) did not think they were vulnerable nor perceive heatwaves as a threat to themselves; They did not understand the increased risks associated with certain medical conditions and medications; Reported behaviours more towards coping rather than mitigation; (B) respondents displayed inconsistent and limited knowledge of heat risks; Also (B) did not want to impinge on (A) independence; Potentially exacerbate (A) vulnerability	Further research into the role of bonding social capital and climate change adaptation Definite need to address barriers in mitigating behaviours Call for government initiatives to finance local social development such as community groups in providing support thus empowering the older people Re-evaluation of adaptation strategies and policy effectiveness
**Herrman et al. 2018 [59]**	**Germany**	Investigate GPs perceptions on susceptibility and nursing care of older people during heatwaves in Baden-Württemberg	n = 24 over four districts	Face-to-face semi-structured interviews, Qualitative software analysed	Exploring knowledge of heatwaves, perceptions of older people morbidity and mortality risks factors and impact levels of future climate change to their well-being	Inconsistent knowledge of heatwaves amongst GPs; Variable levels of concern for older people heat–health based on varied perceptions of risks; Demonstrable uncertainties on impact of climate change on health	More training for GPs on climate change and heatwaves impacts on older people’s health Increase social support and nursing care for older people in extreme weather and heatwaves
**Takahashi et al. 2015 [60]**	**Japan**	Investigate improvement in prevention behaviours and heat health knowledge of older people in Nagasaki	n = 1524 aged 65–84 selected via stratified random sampling	Randomised controlled community trial	Three groups: 1. Heat health warnings + pamphlets 2. Heat health warnings + water bottles + pamphlets 3. Control group	Group 1 took more breaks, reduced activities, wore hats and sun block; Group 2 improved protective behaviours significantly - increased water intake and body cooling; All—poor knowledge of fans usage	Both individual and community based approaches are required for optimal improvement in heat health knowledge and prevention behaviours

HAP: Heat action plan.

**Table 3 ijerph-16-04370-t003:** Ottawa charter action areas.

Author/Year	Build Healthy Public Policy	Create Supportive Environment	Strengthen Community Action	Develop Personal Skills	Reorient Health Services
Hansen et al. 2011 [43]	N/A	N/A	N/A	N/A	**Surveyed healthcare providers and legislators perceptions and knowledge**
Hansen et al. 2015 [44]	N/A	N/A	N/A	**Surveyed heat health behaviours**	N/A
Ibrahim et al. 2012 [45]	N/A	N/A	N/A	N/A	**Investigated health providers knowledge and practices**
McInnes et al. 2010 [46]	N/A	N/A	N/A	N/A	**Investigated community healthcare providers roles**
Nitschke et al. 2013 [47]	N/A	N/A	N/A	**Surveyed risk factors and Protective behaviours**	N/A
Nitschke et al. 2016 [48]	**Evaluation of HWS**	N/A	N/A	N/A	N/A
Nitschke et al. 2017 [49]	**Regional policy in place**	N/A	N/A	**Surveyed protective behaviours**	N/A
**Liotta et al. 2018 [50]**	**National and regional policies in place**	**Coordinated care network**	**Carers and volunteers involved**	**Individuals consent sought**	**Centralised database**
Michelozzi et al. 2010 [51]	**Evaluated national and regional policies**	N/A	N/A	N/A	N/A
Schifano et al. 2012 [52]	**Evaluated efficacy of HAPs**	N/A	N/A	N/A	N/A
Benmarhnia et al. 2016 [53]	**Evaluated effectiveness of HAP**	N/A	N/A	N/A	N/A
Sheridan 2007 [54]	**Reviewed HWS effectiveness**	N/A	N/A	**Surveyed individual responses**	N/A
White-Newsome et al. 2011 [55]	N/A	**Assess effects of indoor temperature**	N/A	**Surveyed protective behaviours**	N/A
Abrahamson et al. 2009 [57]	N/A	N/A	N/A	N/A	**Surveyed health professionals knowledge and perceptions**
Abrahamson et al. 2009 [56]	N/A	N/A	N/A	**Surveyed awareness and behaviours**	N/A
Wolf et al. 2010 [58]	N/A	N/A	N/A	**Surveyed perceptions of vulnerability and behaviours**	N/A
Herrman et al. 2018 [59]	N/A	N/A	N/A	N/A	**Investigated GPs knowledge and practices**
**Takahashi et al. 2015 [60]**	**Regional policy in place**	**Coordinated local network**	**Welfare commissioners involved**	**Surveyed prevention behaviours**	**Overseen by Institute of tropical medicine**

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
