# Peer review of "Heat Health Prevention Measures and Adaptation in Older Populations—A Systematic Review"

_ijerph, 2019, doi:10.3390/ijerph16224370_

Round 1

Reviewer 1 Report

1). Overall : Good paper to advance the conversation of health of an increasingly older population. 

2). Were any efforts made to get any literature representing the developing world? It is important to let the readers know if this was aligned to capturing only developed countries or a global study since the title is general. It is therefore important to probably mention this in the limitations too (and explain if any such studies exist or an area for further research.

3). Line 66 - 67 mentions a 'health framework" but does not explain to the reader what exactly it is until way later in the paper. It is important to explain this as early as possible in the introduction to set the stage because it is a springboard for much of the discussion of the results.

4). Line 334 - 336 introduces the concept of "social determinants of health" but until now this has not been mentioned yet much of what the framework talked about in line 66 - 67 relates and/or is about these factors. It is important to introduce this concept too early in the paper and explain to the reader if this means the same as the Ottawa Charter or not. Then how are they related and why are we talking about both? This will help make some clarifications for readers.

Author Response

Reviewer 1:

Open Review

Comments and Suggestions for Authors

1). Overall: Good paper to advance the conversation of health of an increasingly older population. 

2). Were any efforts made to get any literature representing the developing world? It is important to let the readers know if this was aligned to capturing only developed countries or a global study since the title is general. It is therefore important to probably mention this in the limitations too (and explain if any such studies exist or an area for further research.

We agree with the reviewer that this is a limitation and have actioned on it accordingly. (please see 2 Limitations 444 - 452)

3). Line 66 - 67 mentions a 'health framework" but does not explain to the reader what exactly it is until way later in the paper. It is important to explain this as early as possible in the introduction to set the stage because it is a springboard for much of the discussion of the results.

This has been noted and actioned accordingly. Ottawa Charter action areas moved forwards from 5 Data analysis and synthesis section to 1. Introduction 69 – 117 1.1 Ottawa Charter for Health Promotion Framework with further explanations given.

4). Line 334 - 336 introduces the concept of "social determinants of health" but until now this has not been mentioned yet much of what the framework talked about in line 66 - 67 relates and/or is about these factors. It is important to introduce this concept too early in the paper and explain to the reader if this means the same as the Ottawa Charter or not. Then how are they related and why are we talking about both? This will help make some clarifications for readers.

We have attempted to clarify the social determinants of health further and their associations with the Ottawa Charter (please see 1 The Ottawa Charter for Health Promotion Framework 69 - 84).

Reviewer 2 Report

Comments

General:

The article takes an interesting perspective on heat health impacts for the elderly, which is to look at elderlies’ and health professionals’ awareness and practices of heat health impacts prevention, but also at public health measures such as heat health action plans (HAPs) and heat warning system (HWSs). Furthermore, it analyses the results of the review before the background of the Ottawa Charta, which is an innovative perspective. On the one hand it is a strength of the article to look at individual perspectives on heat health prevention (by elderly and health professionals) before the background of broader public health measures (HAPs, HWSs, Ottawa Charta). On the other hand, in the text it lacks a clear conceptualization of the individual vs. the public health perspective and how they relate to one another. For instance, it would be helpful to define “prevention measures” upstream in the text. It does not become clear throughout the article, whether “prevention measures” mean any kind of individual adaptive behavior, implemented by an individual elderly person or an individual health professional or if it means formalized public health strategies under a HAP or HWS. Also look at comment in the results section.

While the article itself is otherwise quite consistent, the title somewhat mismatches the content. “Heat vulnerability and adaptability in the elderly – a systematic review” suggests that the article reviews studies which look at the epidemiological evidence for increased vulnerability of heat in the elderly. Yet, although the article gives some background information on vulnerability, the content of the review is not about vulnerability. The article certainly looks at adaptation though and takes a focus at 1) measures in heat health action plans and heat warning systems, 2) the perceptions and practices of heat health risks and their prevention in the elderly and health professionals and on 3) effectiveness of implemented measures. Therefore, the title should be adapted to the actual focus of the article (ideally explicitly mention elderly and health staff perceptions, HAPs, maybe also the Ottawa Charta – the things that make the article innovative).

Furthermore, the formulated aims of the article should be streamlined a bit better in the different parts of the article (see a collection of the different description of the aim of the article below).

Abstract: “This systematic review aimed to identify existing heat stress prevention measures in the elderly and systematically categorize and analyse those measures using the Ottawa Charter for health promotion framework.” Introduction: “It aims to contribute to the current body of research on factors influencing elderly people’s adaptive behavior and current preventative strategies during extreme heat events - with two objectives: i) Identify current measures and available evaluations in preventing heat related stress in the elderly, and ii) Systematically categorize and analyse these measures by using a health promotion framework.” Discussion: “The aim of this review was to gather current evidence on enablers and barriers to adaptive behaviors of elderly people against existing mitigating measures in the events of heatwaves.” Conclusions: “The aim of this review was to identify current measures in mitigating the adverse effects of extreme heat; and social and behavioural factors affecting health outcomes of elderly people during ”

Despite these conceptual comments, in summary, the present systematic review has been performed and described in very sound scientific way. It also draws attention to a very urgent issue: while there is enough scientific evidence about heat-related morbidity and mortality, good evidence on how to protect vulnerable populations especially the elderly is indeed lacking. Comparing the current state of public health related prevention measures against the Ottawa Charta is offering a new perspective to address this issue within a broader public health framework.

Abstract:

Contentwise the abstract summarizes the article concisely. The reviewer personally would prefer full sentences instead of the bullet-point: For instance: 15 -16 A genuine lack of systematic evaluation of heat action plans efficacy established. Better: was identified. 19-20: Governments beginning to enact area 1 of the Ottawa Charter but gaps are still evident in all areas. Better: Governments are beginning In the Methodology part it would be nicer of the systematic literature search was shortly described 12-13: Peer-reviewed published literature between 22nd September 2006 and 24th April 2018 were utilized. Better: were retrieved according to the PRISMA checklist from five different databases. The reviewer couldn’t see the finding, that only area 1 of the Charta is enacted by all, while others are enacted by some in the full text. 19 -20: Governments beginning to enact area 1 of the Ottawa Charter but gaps are still evident in all areas.

Introduction

The introduction gives the right background information to understand the need for the study.

Methodology:

The methodology is implemented scientifically correct and is well described.

Results

The results are well described and the table is well structured.

Correlating to my previous comment on “preventive measures (whether these mean individual or formalized public health measures), it would be helpful to conceptualize, why the article does not only look at HAP measures and their effectiveness (3.2 and 3.4.), but also explicitly focuses on “ 3.3 Risk factors awareness, Perception of vulnerability and Protective behaviors” of elderly AND health professionals together. Health professionals are certainly official stakeholders in heat health action plans, so that their “awareness, perceptions and protective behaviours” could be seen as an element of evaluating the effective implementation of HAPs. Elderly are rather the target population of HAPs. So why is that the article also looks at their “awareness, perceptions and protective behaviours”? Is it in order to identify the need for HAPs or is it a kind of evaluation to see whether HAPs have been effective? It does make sense, but it should explicitly conceptualized why. Furthermore, it might conceptually make sense to look elderly and health professionals separately (no absolute need, depending on the final concept).  

Discussion:

One could also think about including the analysis within the Ottawac charta framework into the results, but it is also fine to keep it here. “228 The aim of this review was to gather current evidence on enablers and barriers to adaptive behaviors of elderly people against existing mitigating measures in the events of heatwaves.” Personally, I don’t think that enablers and barriers really catches correctly what has been done in the review. I would rather talk about “3.3 Risk factors awareness, Perception of vulnerability and Protective behaviors”. Furthermore, the mentioned results subchapter pertains to elderly and health professionals, so maybe this should also be mentioned here?

Conclusions

Depending on how the role of health professionals and elderly has been conceptualized in the final article, one could state here, that one component of evaluating public health programs (knowledge and practices of involved health professionals) has already been addressed in several studies, but that broader evaluations, especially quantitative and qualitative ones are missing.

Otherwise well drawn conclusions.

Some more articles of potential interest:

Bittner, M. I., & Stößel, U. (2012). Perceptions of heatwave risks to health: results of an qualitative interview study with older people and their carers in Freiburg, Germany. Psychosoc Med, 9. doi: doi:  10.3205/psm000083

van Loenhout, J. A., Rodriguez-Llanes, J. M., & Guha-Sapir, D. (2016). Stakeholders' Perception on National Heatwave Plans and Their Local Implementation in Belgium and The Netherlands. Int J Environ Res Public Health, 13(11). doi:10.3390/ijerph13111120

 Ragettli MS, Vicedo-Cabrera AM, Schindler C, Röösli M (2017) Exploring the association between heat and mortality in Switzerland between 1995 and 2013. Environ Res 158:703–709.  https://doi.org/10.1016/j.envres.2017.07.021

Author Response

Reviewer 2:

General:

The article takes an interesting perspective on heat health impacts for the elderly, which is to look at elderlies’ and health professionals’ awareness and practices of heat health impacts prevention, but also at public health measures such as heat health action plans (HAPs) and heat warning system (HWSs). Furthermore, it analyses the results of the review before the background of the Ottawa Charta, which is an innovative perspective. On the one hand it is a strength of the article to look at individual perspectives on heat health prevention (by elderly and health professionals) before the background of broader public health measures (HAPs, HWSs, Ottawa Charta). On the other hand, in the text it lacks a clear conceptualization of the individual vs. the public health perspective and how they relate to one another. For instance, it would be helpful to define “prevention measures” upstream in the text. It does not become clear throughout the article, whether “prevention measures” mean any kind of individual adaptive behavior, implemented by an individual elderly person or an individual health professional or if it means formalized public health strategies under a HAP or HWS. Also look at comment in the results section.

We have taken this comment on board and clarified the definition of prevention measures in the context of this systematic review (please see 72 -77).

While the article itself is otherwise quite consistent, the title somewhat mismatches the content. “Heat vulnerability and adaptability in the elderly – a systematic review” suggests that the article reviews studies which look at the epidemiological evidence for increased vulnerability of heat in the elderly. Yet, although the article gives some background information on vulnerability, the content of the review is not about vulnerability. The article certainly looks at adaptation though and takes a focus at 1) measures in heat health action plans and heat warning systems, 2) the perceptions and practices of heat health risks and their prevention in the elderly and health professionals and on 3) effectiveness of implemented measures. Therefore, the title should be adapted to the actual focus of the article (ideally explicitly mention elderly and health staff perceptions, HAPs, maybe also the Ottawa Charta – the things that make the article innovative).

We have changed the title in accordance with this observation (please see 2 – 3).

Furthermore, the formulated aims of the article should be streamlined a bit better in the different parts of the article (see a collection of the different description of the aim of the article below).

Abstract: “This systematic review aimed to identify existing heat stress prevention measures in the elderly and systematically categorize and analyse those measures using the Ottawa Charter for health promotion framework.” Introduction: “It aims to contribute to the current body of research on factors influencing elderly people’s adaptive behavior and current preventative strategies during extreme heat events - with two objectives: i) Identify current measures and available evaluations in preventing heat related stress in the elderly, and ii) Systematically categorize and analyse these measures by using a health promotion framework.” Discussion: “The aim of this review was to gather current evidence on enablers and barriers to adaptive behaviors of elderly people against existing mitigating measures in the events of heatwaves.” Conclusions: “The aim of this review was to identify current measures in mitigating the adverse effects of extreme heat; and social and behavioural factors affecting health outcomes of elderly people during ”

We have taken this comment on board and made an effort to streamline the aims throughout as suggested by the reviewer.

Despite these conceptual comments, in summary, the present systematic review has been performed and described in very sound scientific way. It also draws attention to a very urgent issue: while there is enough scientific evidence about heat-related morbidity and mortality, good evidence on how to protect vulnerable populations especially the elderly is indeed lacking. Comparing the current state of public health related prevention measures against the Ottawa Charta is offering a new perspective to address this issue within a broader public health framework.

Abstract:

Contentwise the abstract summarizes the article concisely. The reviewer personally would prefer full sentences instead of the bullet-point: For instance: 15 -16 A genuine lack of systematic evaluation of heat action plans efficacy established. Better: …was identified. 19-20: Governments beginning to enact area 1 of the Ottawa Charter but gaps are still evident in all areas. Better: Governments are beginning… In the Methodology part it would be nicer of the systematic literature search was shortly described 12-13: Peer-reviewed published literature between 22nd September 2006 and 24th April 2018 were utilized. Better: …were retrieved according to the PRISMA checklist from five different databases. The reviewer couldn’t see the finding, that only area 1 of the Charta is enacted by all, while others are enacted by some in the full text. 19 -20: Governments beginning to enact area 1 of the Ottawa Charter but gaps are still evident in all areas.

We have taken this comment on board and actioned accordingly (please see 9 - 24)

Introduction

The introduction gives the right background information to understand the need for the study.

Methodology:

The methodology is implemented scientifically correct and is well described.

Results

The results are well described and the table is well structured.

Correlating to my previous comment on “preventive measures (whether these mean individual or formalized public health measures), it would be helpful to conceptualize, why the article does not only look at HAP measures and their effectiveness (3.2 and 3.4.), but also explicitly focuses on “ 3.3 Risk factors awareness, Perception of vulnerability and Protective behaviors” of elderly AND health professionals together. Health professionals are certainly official stakeholders in heat health action plans, so that their “awareness, perceptions and protective behaviours” could be seen as an element of evaluating the effective implementation of HAPs. Elderly are rather the target population of HAPs. So why is that the article also looks at their “awareness, perceptions and protective behaviours”? Is it in order to identify the need for HAPs or is it a kind of evaluation to see whether HAPs have been effective? It does make sense, but it should explicitly conceptualized why. Furthermore, it might conceptually make sense to look elderly and health professionals separately (no absolute need, depending on the final concept). 

We have taken this comment on board and made an effort to clarify ‘prevention measures’ in the context of this systematic review. We understood ‘prevention’ as actions taken by the individuals, as well as the local communities and federal agencies as specified by regional legislations and policies – hence the use of the Ottawa Charter as a framework.  It is indeed essential to examine older people as target populations but also their thoughts and behaviours so as to inform implementation and/or refinement of policies and procedures (HWS/HAPs) catered for such populations. (e.g. Benmarhnia, T.; Bailey, Z.; Kaiser, D.; Auger, N.; King, N.; Kaufman, J. S. A Difference-in-Differences Approach to Assess the Effect of a Heat Action Plan on Heat-Related Mortality, and Differences in Effectiveness According to Sex, Age, and Socioeconomic Status (Montreal, Quebec). Environmental Health Perspectives. 2016, 124, 1694-1699.

Appendix 1: Examples of Actions Pertaining to the Montreal Heat Action Plan

Transmission of information coming from the Montreal PHD concerning levels of the plan that must be implemented in the health care network. Intensification of surveillance of signs and symptoms of heat-related illness, reminder of preventive measures to patients, distribution of water bottles. Air conditioning of common areas and opening of these areas during the day, the evening, and the night for patients in institutions. Frequent visits to home care patients. In institutions, frequent visits to housed patients. Distribution of water, refreshments, lighter meals. Monitoring of temperature in work areas, especially in warmer environments (e.g., kitchen, laundry room). Frequent work breaks for workers in hot, non-air-conditioned environments. Transfer of patients to common areas with air conditioning. Transfer of vulnerable home care patients to air-conditioned shelters. Daily contact by telephone or home visits to home care patients. Registry of calls and compilation of questionnaires for home evaluation.)

Discussion:

One could also think about including the analysis within the Ottawac charta framework into the results, but it is also fine to keep it here. “228 The aim of this review was to gather current evidence on enablers and barriers to adaptive behaviors of elderly people against existing mitigating measures in the events of heatwaves.” Personally, I don’t think that enablers and barriers really catches correctly what has been done in the review. I would rather talk about “3.3 Risk factors awareness, Perception of vulnerability and Protective behaviors”. Furthermore, the mentioned results subchapter pertains to elderly and health professionals, so maybe this should also be mentioned here?

We have taken this comment on board and made changes accordingly (please see 277 – 280).

Conclusions

Depending on how the role of health professionals and elderly has been conceptualized in the final article, one could state here, that one component of evaluating public health programs (knowledge and practices of involved health professionals) has already been addressed in several studies, but that broader evaluations, especially quantitative and qualitative ones are missing.

Otherwise well drawn conclusions.

We have made an effort to improve upon the conclusions (please see 459).

Some more articles of potential interest:

Bittner, M. I., & Stößel, U. (2012). Perceptions of heatwave risks to health: results of an qualitative interview study with older people and their carers in Freiburg, Germany. Psychosoc Med, 9. doi: doi:  10.3205/psm000083

van Loenhout, J. A., Rodriguez-Llanes, J. M., & Guha-Sapir, D. (2016). Stakeholders' Perception on National Heatwave Plans and Their Local Implementation in Belgium and The Netherlands. Int J Environ Res Public Health, 13(11). doi:10.3390/ijerph13111120

 Ragettli MS, Vicedo-Cabrera AM, Schindler C, Röösli M (2017) Exploring the association between heat and mortality in Switzerland between 1995 and 2013. Environ Res 158:703–709.  https://doi.org/10.1016/j.envres.2017.07.021

We thank the reviewer for these articles and have made an effort to incorporate Bittner et al. (2012) and van Loenhout et al. (2016) into our discussion (please see 303 – 305 and 406)

Author Response

Reviewer 3:

ijerph-621601

 --Abstract could be improved; having incomplete sentences does not help to build a case for this work. I would use full sentences and refine. E.g., “Governments beginning to enact area 1 of the Ottawa Charter but gaps are still evident in all areas” is quite unclear. What is area 1? What kinds of gaps are still evident? I advise that authors use in abstract only information that is complete and perhaps add some clarification what Ottawa charter is considering it serves as a framework for the assessment.

This comment has been taken on board and actioned accordingly.

--There is a big debate on the use of word “elderly” and it seems that most scholars agree that this word is not appropriate to describe this subpopulation  (for example, please see: https://www.tandfonline.com/doi/full/10.1080/01634372.2015.1033363)

We agree with the reviewer that the word ‘elderly’ conjures negative connotations pertaining to the aging process and thus somewhat diminishes the person behind that process. However, we must point to the fact that older people are regarded differently in different cultures and hence represented by different terms depending on cultural contexts.  One immediate example is that of the Long Live the Elderly program as described by Liotta et al’s study. Here the authors have used the word ‘elderly’ in a positive empowering context.  Therefore, to capture all appropriate articles we would have to use terms that have been employed by authors in all contexts and cultures (e.g. references [10,14,15,17,22,31,36,52,59,60]).  However, we shall make every effort to use respectful language from here on in and thus replaced all ‘elderly’ words with ‘older people’ in our systematic review.

 --My concern is that search based on criteria “(Elderly OR older people OR seniors OR “vulnerable populations” OR “vulnerable groups”)” didn’t yield all papers because of my comment above. Also considering very specific key search combo “older people” it may not pick up on papers that were looking at “older population” or “Older adults”. Search using only “older” would probably fix that. Also, inclusion of “vulnerable groups” is peculiar considering how many vulnerable social groups there are. 

We have taken on board this feedback and repeated searches through the same databases with the following keywords: (Heat OR heat-related OR “high temperatures” OR heatwaves) AND (Elderly OR older OR senior* OR “vulnerable populations” OR “vulnerable groups”) AND (Management OR “preventative measures” OR prevention OR solutions OR mitigation OR adaptation OR intervention OR evaluation) NOT (Occupation OR work). The results were then combined with our previous library in Endnote.  Duplicates were removed – meaning all articles that were previously identified from the original keywords and this most recent attempt using new keywords above (inception to July 2019) were deleted.  The remaining articles were scanned for inclusion as per our original criteria and none of them fulfilled the requirements. Therefore, we have concluded that removing the word ‘people’ after the word ‘older’ made no difference to the outcome. Also, please note that we did not use “older people” but rather older people without the “” and evidently we have found articles with various versions of the word ‘older’ including ‘older population’ and ‘older adults’ (e.g. references [49,76,79,85]) even prior to repeating the searches. We have justified the use of the word ‘elderly’ in the keywords (please see above). In addition, in the area of research on climate change and indeed extreme temperatures, most authors would include older people and children, amongst certain groups such as low socio-economic status or ethnic/migrants as ‘vulnerable populations’ or ‘vulnerable individuals’. One example of authors who used the word ‘vulnerable’ groups to describe people who are 65 years and older and/or in poor health: “Preventive measures deployed as part of the heat plan apply to the entire population, although some actions are focused towards individuals that may be more vulnerable to heat. In the Montreal Heat Response Plan, vulnerable individuals include the elderly and those with certain chronic physical or mental illness, in particular when there is no access to air conditioning in the home or when individuals are living alone or are socially isolated.” (Price, K.; Benmarhnia, T.; Gaudet, J.; Kaiser, D.; Sadoine, M. L.; Perron, S.; Smargiassi, A. The Montreal heat response plan: evaluation of its implementation towards healthcare professionals and vulnerable populations. Canadian Journal of Public Health-Revue Canadienne De Sante Publique. 2018, p109, 108-116.). Hence, leaving those keywords out would actually be counterproductive to our search strategy.  Another example is: “Those with higher sensitivity include women, children, the elderly, and those with chronic illnesses and disability. Those in urban heat islands, floodplains, and coastal communities are at greater exposure, and socioeconomically disadvantaged groups may have both greater exposure and less capacity for adaptation” (Akerlof, K. L., Delamater, P. L., Boules, C. R., Upperman, C. R., & Mitchell, C. S. (2015). Vulnerable populations perceive their health as at risk from climate change. International Journal of Environmental Research and Public Health, 12(12), 15419-15433. doi:10.3390/ijerph121214994).

 --The protocol is generally accurate but it is surprising to start with 7133 articles and end up with only 18 (or 620 to 30). I am wondering if criteria are too specific. This is a very small sample and again, I am not sure whether this qualifies.

We refer to two reliable sources in which neither specified the minimum number of articles to be included in a systematic literature review. In addition, we have made every effort to follow the PRISMA guidelines as specified in section Methods. Gough, D. (. A. )., Oliver, S., 1955, & Thomas, J., 1983. (2012). An introduction to systematic reviews. London: Sage. Shamseer, L., Moher, D., Clarke, M., Ghersi, D., Liberati, A., Petticrew, M., . . . the PRISMA-P Group. (2015). Preferred reporting items for systematic review and meta-analysis protocols (PRISMA-P) 2015: Elaboration and explanation. BMJ : British Medical Journal, 349(jan02 1), g7647-g7647. doi:10.1136/bmj.g7647. We have deliberately not filtered out clinical journals despite the topic being a public health issue in order to capture the odd article that may have been published in medical journals. Evidently, there is an overwhelming focus on clinical trials and treatment focused studies rather than preventative public health focused studies. Furthermore, it highlights the fact that research in this area (climate change adaptation in older populations) is scarce and hence strengthens our purpose and adds to the urgency to explore this topic further.

 --Considering none of the 18 papers had evaluation, this column can be removed from the table and this can be noted in the text. 

This has been noted and actioned accordingly.

--I also suggest that authors refer to specific studies that are emphasized in Results section (e.g., some studies describe them in more detail (e.g. Smith 2018, Jones 2019)

This has been noted and actioned accordingly.

--The section on the Ottawa Charter Action Areas could be better integrated and connected with the previous sections. It lists relevance of action areas but could do a better job in articulating their application considering the findings of systematic review. Some paragraphs do a great job e.g., 4.1.4 Develop personal skills.

We have made an effort to improve on this by defining and explaining the Ottawa Charter in more details (please see Introduction 69 - 117)

--As of study characteristics, in addition to describing different approaches, I would love to see a more rigorous discussion of different approaches and context of individual studies.

We have attempted to clarify the usefulness of various study designs in health research in relation to included studies limitation (please see 2 Limitations 431 – 443). We would like to invite further research and discussions in this area as outlined in our conclusions.

Round 2

Reviewer 3 Report

Thank you for revisions and response to my comments.